# The Role of Exosomes in Inflammatory Diseases and Tumor-Related Inflammation

**DOI:** 10.3390/cells11061005

**Published:** 2022-03-16

**Authors:** Yuan Tian, Cheng Cheng, Yuchong Wei, Fang Yang, Guiying Li

**Affiliations:** 1Key Laboratory for Molecular Enzymology and Engineering of the Ministry of Education, School of Life Sciences, Jilin University, Changchun 130012, China; tianyuan@ccucm.edu.cn (Y.T.); chengcheng20@mails.jlu.edu.cn (C.C.); weiyc1319@mails.jlu.edu.cn (Y.W.); fangyang@jlu.edu.cn (F.Y.); 2Department of Pathology and Pathophysiology, Clinical School of Medicine, Changchun University of Traditional Chinese Medicine, Changchun 130117, China

**Keywords:** exosomes, inflammatory disease, tumor microenvironment, tumor-related inflammation

## Abstract

Inflammation plays a decisive role in inducing tumorigenesis, promoting tumor development, tumor invasion and migration. The interaction of cancer cells with their surrounding stromal cells and inflammatory cells further forms an inflammatory tumor microenvironment (TME). The large number of cells present within the TME, such as mesenchymal stem cells (MSCs), macrophages, neutrophils, etc., play different roles in the changing TME. Exosomes, extracellular vesicles released by various types of cells, participate in a variety of inflammatory diseases and tumor-related inflammation. As an important communication medium between cells, exosomes continuously regulate the inflammatory microenvironment. In this review, we focused on the role of exosomes in inflammatory diseases and tumor-related inflammation. In addition, we also summarized the functions of exosomes released by various cells in inflammatory diseases and in the TME during the transformation of inflammatory diseases to tumors. We discussed in depth the potential of exosomes as targets and tools to treat inflammatory diseases and tumor-related inflammation.

## 1. Introduction

Years ago, cancer was thought to be a cell-autonomous process. However, an increasing number of studies on tumor microenvironment (TME) have provided a new explanation for the occurrence and development of tumors; that is, tumor development depends on the complex interaction between tumor cells, cells in the surrounding environment and the vasculature [1,2]. Inflammation is a defense-dominated local tissue response to stimulation by pathogens, damaged cells, or irritants, usually manifested by redness, swelling, heat, pain and dysfunction [3,4]. A striking feature of inflammation is that it has core cells and molecules required for tumorigenesis [5]. In preclinical animal models, inhibition of inflammation also suppressed tumor growth and progression in certain seemingly “non-inflammatory” cancers [6]. Although the mechanism by which inflammatory cells lead to tumorigenesis is still unclear, recent studies on tumorigenesis have shown that inflammation is likely to induce early tumorigenesis [7,8]. Inflammation-promoting carcinogenesis may be the result of a changing environment interacting with a variety of cells, such as increased genomic instability, abnormal cell proliferation, changes in the stromal environment and transitions between epithelial and mesenchymal states [9,10]. Inflammatory factors can activate inflammation-related transcription factors, leading to the activation of pro-tumor signaling pathways, so inflammation may induce tumorigenesis [11,12,13]. With the accumulation of inflammation, it may further promote tumor development. In the later stages of tumor development, the inflammatory environment easily leads tumor cells to undergo an epithelial-mesenchymal transition (EMT) process, and on the other hand, with the help of tumor-associated macrophages (TAMs), enter blood vessels and lymphatic vessels, thereby achieving a transfer process [14,15]. Inflammation usually resolves in the short term, but chronic or persistent inflammation may lead to malignant disease [16,17]. Epidemiological data have demonstrated a relationship between chronic inflammation and the development of several cancers, including liver cancer, lung cancer and colorectal cancer [18].

Exosomes, small extracellular vesicles with an average size of ~100 nm, are released by various types of cells [19]. In recent years, several research groups have reported the potential biological functions of exosomes secreted by different types of cells, which play an important role in maintaining physiological homeostasis. During tumor development, the composition of exosomes undergoes various changes, which play a role in promoting the progression of various tumors [2,20,21]. The specific mechanisms include promoting cell proliferation, promoting angiogenesis, promoting cancer cell metastasis and invasion, reprogramming metabolic processes, and modulating the TME to cause immune evasion, etc. [1,22]. A large number of studies about exosomes in various inflammatory diseases have been reported, and exosomes have also become potential therapeutic targets for various chronic inflammatory diseases and even tumors [23,24]. In this review, we systematically summarized the roles of exosomes in various inflammatory diseases and during the transformation of inflammatory diseases to tumors. In addition, we also summarized the functions of exosomes released by various cells in inflammatory diseases. The potential of exosomes as a target for the treatment of inflammatory diseases and inflammation-related tumors is discussed in depth.

## 2. Exosomes Composition and Function

Exosomes were originally thought to be small vesicles containing 5′ nucleotidase activity released from tumor cell lines [25]. A few years later, Johnstone’s group reported mature mammalian reticulocyte-derived vesicles (30–120 nm in diameter) that reside in large multicellular vesicles containing transferrin receptors and several other membrane-associated vesicles [26]. Using electron microscopy, it was observed that fusion of multivesicular endosomes with the plasma membrane could release inner vesicles into the extracellular environment [27]. A large number of subsequent studies have found that exosomes could be released by various types of cells including mesenchymal stem cells (MSCs), macrophages, neutrophils, dendritic cells (DC) and lymphocytes, also be found in most body fluids including blood, urine, and saliva [19]. With the deepening of research on exosomes, more and more evidence showed that exosomes played an important role in the material exchange process between cells, and most of the physiological and pathological processes of the body [28,29]. In addition, studies have found that the molecular components within exosomes were significantly correlated with certain diseases, suggesting that they could also serve as a diagnostic tool [30,31].

Exosomes have unique and complex compositions. Exosomes mainly include a variety of proteins, lipids, mRNAs and miRNAs [19]. The most common proteins found in exosomes are membrane transport and fusion-related proteins (such as GTPase, annexin), heat shock proteins (such as HSC70), tetraspanin superprotein family (such as CD63 and CD81), multivesicular body (MVB) synthesis proteins (such as TSG101), cholesterol, phospholipids and other lipids [32,33]. CD63 and CD81 are the most commonly used surface marker molecules for the detection of exosomes [34]. In addition, exosomes are also found to contain a large amount of mRNA and miRNA, which have different effects on recipient cells [35,36].

Exosomes can fuse with recipient cells and release their contents, transferring components from secretory cells to target cells, so exosomes play an important role in cell-to-cell communication. Certain immune cell-derived exosomes, such as exosomes released by DC cells and B cells, could mediate the body’s adaptive immune response to pathogens and tumors [37]. Tumor cell-derived exosomes could promote tumor development and metastasis [38]. Exosomes shed from blood cells and vascular endothelium can be involved in neurological diseases such as multiple sclerosis, transient ischemia, and antiphospholipid syndrome [39].

## 3. The Role of Exosomes in Inflammatory Diseases

### 3.1. Exosomes in Sepsis Associated Inflammation

Sepsis is a life-threatening systemic inflammatory disease caused by bacterial and other pathogenic microorganisms invading the body [40]. Sepsis occurs when the body is simultaneously unbalanced with excessive inflammation and immunosuppression [41]. Numerous literature reports on the clinical importance of exosomes as biomarkers and mediators in sepsis. Exosomes can induce inflammation by releasing their contents, thereby activating receptor cells, and experimental clinical data showed that the release of exosomes were significantly increased when sepsis occurred [42]. The exosomes detected in the early stage of septic mice contain a large number of pro-inflammatory factors, such as TNF-α, IL-6, etc., while the anti-inflammatory factor IL-10 can be detected in the late stage [43]. One study found that exosomes carrying a large number of damage-associated molecular patterns (DAMPs) were released by secretory cells during sepsis, including high mobility group box 1 (HMGB1) [44,45], heat shock protein, adenosine triphosphate (ATP) and extracellular RNA, etc. [46]. These DAMP molecules can bind to pattern recognition receptors (PRRs), especially Toll-like receptors (TLRs), to initiate inflammatory signaling [47]. The most well-known mechanism of sepsis is that lipopolysaccharide (LPS) activates the TLR4-MyD88 pathway to activate the downstream NF-κB signaling pathway, resulting in the production of a large number of inflammatory molecules [47]. In septic mice, exosomes were found to activate downstream signaling through TLRs, promoting the production of cytokines and chemokines [48,49]. NLRP3 inflammasome, a typical Nod-like receptor, is also an important PRR in sepsis, and it has been reported that exosomes released from LPS-treated macrophages could induce NLRP3 inflammasome and caspase-1 activates and induces the release of the proinflammatory factor IL-1β [50,51]. Therefore, exosomal inflammasomes may be involved in aggravating inflammation in sepsis.

### 3.2. Exosomes in Lung Inflammatory Disorders

At present, the morbidity and mortality of inflammatory lung disease are still high, and the hallmark of many respiratory diseases is the production of inflammation [52]. The lungs have abundant blood vessels, and the exosomes released from lung endothelial cells contain a large number of membranous mucins, which help the innate defense of the airway [53]. At the same time, exosomes have attracted attention in various inflammatory lung diseases, such as chronic obstructive pulmonary disease (COPD), acute lung injury (ALI), asthma, and COVID-19. COPD is a lung disease caused by endothelial cell damage, epithelial cell damage, and epithelial-mesenchymal transition in the lung parenchyma [54]. It has been reported that exosomes released by lung endothelial cells are significantly increased after infection and smoke exposure [55]. These exosomes were found to induce increased IL-8 secretion, causing lung tissue damage and persistent inflammation in COPD lungs [56]. In addition, lung endothelial cells release a large number of exosomes into bronchoalveolar lavage fluid (BALF) after injury, which affects the function of recipient cells. Therefore, exosomes in BALF can also be used to predict the extent of COPD damage [57]. The study by Lee et al. (2018) showed that the sources of exosomes in ALI and BALF of acute respiratory distress syndrome (ARDS) were mainly alveolar type I epithelial cells and alveolar macrophages [58]. Exosomes from BALF of mice with LPS-induced ALI contain a large amount of cytokines and Caspase-1, which induce apoptosis of lung endothelial cells and damage the alveolar-capillary barrier [59]. It was also observed in asthma model mice that lung epithelial cells secreted more exosomes, became the main source of lung exosomes, and induced the proliferation and chemotaxis of monocytes, while inhibiting the secretion of exosomes could relieve asthma symptoms [60]. Asthma is also often accompanied by increased eosinophils in the airways and production of more exosomes that promote the inflammatory behavior of eosinophils associated with asthma pathogenesis [61,62]. In addition, exosomes secreted by mast cells of asthmatic mice carry molecules such as MHC class II and ICAM-1, which can induce the activation and recruitment of splenic B cells and T cells, leading to lung inflammation [63]. COVID-19, which is now spreading globally, has caused severe inflammation in the lungs [64]. Proteomics analysis of exosomes derived from COVID-19 patients revealed that some molecules involved in inflammatory responses, such as complement C1r and C1s subcomponents, can be used as potential biomarkers [65]. Because of their unique lipid bilayer-enclosed structure and function of intercellular communication, exosomes may be used in the development of antiviral drugs and vaccines. In addition, in lung cancer, exosomes can still be observed to promote tumor development. Abundant research shows that tumor-derived exosomes can regulate local immune responses, epithelial-mesenchymal transition (EMT), angiogenesis and other pathways, thereby promoting lung cancer cell proliferation, migration and invasion [66,67].

### 3.3. Exosomes in Liver Inflammation

Additionally, of high concern is liver injury. In most liver diseases, liver injury triggers the death of liver cells, which in turn leads to liver failure, liver fibrosis, and hepatocellular carcinoma [68]. In this process, there is always a long, hyperactive inflammatory response. Inflammation in the initial stage of liver injury plays a role in tissue repair, but excessive inflammation over time lead to liver cell damage and death [69]. Researches have shown that after hepatocyte injury, DAMP molecules were packaged into exosomes, causing non-parenchymal cells to synthesize and release pro-inflammatory cytokines, such as IL-1β and TNF-α, leading to local inflammation [70]. The exosomes released from hepatocytes further promote the entry of immune cells, such as macrophages, into the liver, thereby maintaining and amplifying inflammation [71]. However, some studies have also shown that acetaminophen (APAP)-induced acute liver injury reached the most serious injury at 24 h, then gradually repaired, and basically returned to normal at 72 h. During this period, resident and infiltrating macrophages in the liver play a key role in the process of tissue injury repair [72]. Exosomes also transmit signals to endothelial cells, leading to inflammation of blood vessels. In the study of viral hepatitis, exosomes can regulate the host immune response and mediate hepatitis virus replication. Exosomes released by hepatitis virus-infected hepatocytes help hepatitis virus to participate in immune escape [73]. Of course, they can also activate the body’s immune response to infection with hepatitis virus, help eliminate the virus and activate the antiviral immune response [74]. Not only that, more and more studies have shown that exosomes also played an important role in promoting liver fibrosis. Liver fibrosis is a dynamic process in which excessive inflammation directly or indirectly drives the activation of hepatic stellate cells (HSCs) [75]. Previous studies have shown that lipid-induced hepatocyte-derived exosomes regulate HSCs activation by delivering miR-128-3p and inhibit PPAR-γ expression, resulting in a significant increase in profibrotic gene expression [76]. It has also been demonstrated that exosomes produced by damaged epithelial cells promoted increased production of α-smooth muscle actin and type I collagen in HSCs [77]. Furthermore, platelet-derived growth factor (PDGF)-activated hematopoietic stem cells release PDGFRa-enriched exosomes that induce HSC migration and liver fibrosis [78]. These findings suggest that exosomes may be key regulators of liver fibrosis.

Under conditions of liver injury, the inflammatory environment leads to hepatocyte necrosis and chromosomal instability. Reactive oxygen species and inflammatory factors may trigger the occurrence of hepatocellular carcinoma (HCC) [75]. During the progression of HCC, exosomes that play a role in promoting tumor development gradually increased. These exosomes promote angiogenesis, EMT, matrix remodeling and immune regulation, thereby promoting tumor progression and metastasis [79]. It has been reported that exosomes were involved in the development of chronic hepatitis B (CHB) and chronic hepatitis C (CHC) into HCC [80]. The exosomes released by the transferred HCC cells in turn carried a large amount of tumorigenic RNAs and proteins, enhanced the phosphorylation of PI3K/AKT and MAPK pathways and the production of matrix metalloproteinase (MMP)-2 and MMP-9, which promoted the migration and invasion capacity of MIHA cells [81]. In addition, HCC cell-derived exosomes evade immune surveillance by inducing tumor-infiltrating NK cell dysfunction by activating the TGF-β/Smad pathway [82].

### 3.4. Exosomes in Inflammatory Bowel Diseases

Inflammatory bowel disease (IBD) is an idiopathic intestinal inflammatory disease caused by long-term inflammation coupled with immune dysregulation leading to damage to the gastrointestinal tract, including ulcerative colitis (UC) and Crohn’s disease (CD) [83]. In the IBD microenvironment, exosomes play a role in regulating immune cells and gut microbiota. It has been reported that compared with exosomes from healthy subjects, exosomes isolated from the colonic lumen of IBD patients contained more inflammatory factors such as IL-6 and TNF-α, and the expression levels of these pro-inflammatory molecules increased with the severity of CD [84]. In addition, it was found that exosomes from IBD patients were able to induce the activation of colonic epithelial cells in vitro to produce IL-8 [84]. Intestinal epithelial cells treated with exosomes from IBD patients were able to induce higher numbers of macrophage recruitment than untreated intestinal epithelial cells [85]. Studies have shown that compared with control mice, mice with colitis induced by dextran sodium sulfate (DSS) contained 56 differential proteins in serum exosomes, a large number of which could induce macrophages to release more TNF-α [86]. In addition, it has also been reported that the 1963 proteins in milk-derived exosomes, including Flotillin-1, Annexin A5, were involved in the regulation of the gut microbiome in the murine IBD mucosal microenvironment [87]. Other studies have reported that exosomes from different types of milk could enhance intestinal epithelial cell (IEC) activity, proliferative capacity and intestinal development [88]. In addition, exosomes released by IECs present exogenous peptides to T cells after activating DCs, which further activates the inflammatory response [89]. Another study found that after infection with adherent invasive *Escherichia coli* (AIEC), intestinal epithelial cells and macrophages secreted increased exosomes and led to an enhanced pro-inflammatory response [90]. These all indicate that the increased release of exosomes caused by inflammation may further promote the infiltration of inflammation in the intestine to varying degrees. IBD has long been associated with an increasing risk of colorectal cancer. In the process of colitis-associated colorectal cancer (CRC), the activation of STAT3 by IL-6 is the key to early tumorigenesis. STAT3 promotes tumor cells by up-regulating the expression of cell cycle regulators cyclin D1, cyclin D2 and the proto-oncogene MYC [91]. In addition, miR-1246 in colon cancer cell-derived exosomes is a key mediator in promoting the transformation of macrophages into a tumor-promoting phenotype [92]. Additionally, CRC-derived exosomes up-regulated the expression levels of VEGF, Wnt5A, and IL-1β, leading to TAMs differentiation [93]. In addition to this, exosomes also contribute to colon cancer metastasis. The Wnt1 protein in exosomes was shown to promote the proliferation and migration of CRC cells [94]. It was also found that miR-424-5p in exosomes of CRC cells promoted the proliferation and metastasis of colorectal cancer by directly inhibiting the tumor suppressor gene SCN4B [95]. Furthermore, exosomes from CRC cells induced a malignant phenotype when injected into normal colon cells in vitro [96]. All of the above prove that exosomes play an important role in the development of inflammatory bowel disease and related tumors.

## 4. Role of Exosomes Released by Microenvironmental Cells in Inflammatory Diseases and Tumor-Related Inflammation

### 4.1. Mesenchymal Stem Cells-Derived Exosomes

Several studies have shown that mesenchymal stem cell (MSCs)-derived exosomes, especially the miRNAs in them, played a key role in animal models of sepsis. For example, miRNA-141 in MSC-derived exosomes can alleviate myocardial injury in septic mice after activation of β-catenin through PTEN [97]. Another report found that miRNA-21 was massively upregulated in exosomes released by IL-1β-stimulated MSCs, which induced macrophage polarization to M2 type and thus ameliorated sepsis [98]. The up-regulation of miR-223 in exosomes leads to the downregulation of Sema3A and STAT3, indicating that miR-223 in exosomes plays an important role in cardioprotection of sepsis [99].

In addition, some preclinical studies have also demonstrated that the main effects of MSC-derived exosomes on ALI/ARDS were to reduce lung inflammation, promote alveolar epithelial regeneration, restore alveolar fluid clearance and enhance pulmonary endothelial repair [100]. In addition, MSC-derived exosomes can modulate the immune response in ALI/ARDS [101]. MSC-derived exosomes induce M1-type macrophage polarization to M2-type by transferring miRNA-27a-3p to alveolar macrophages [102]. MSC-derived exosomes can repair the intestinal barrier in various ways. For example, miRNA-181a in MSC-derived exosomes can improve the state of experimental colitis by promoting intestinal barrier and anti-inflammatory effects [103]. Human umbilical cord MSC-derived exosomes repair intestinal barrier through tumor necrosis factor-α-stimulated gene 6 (TSG-6) for the treatment of inflammatory bowel disease [104]. In addition, miRNA-34/c-5p and miRNA-29-3p in MSC-derived exosomes can improve the intestinal epithelial barrier through Snail/Claudins signaling pathway [105].

In mouse model experiments, it was observed that exosomes derived from human bone marrow mesenchymal stem cells (hBMSCs) could promote angiogenesis by upregulating the expression of VEGF and promote tumor cell proliferation by activating ERK1/2 and p38 MAPK pathways, leading to the development of SGC-7901 gastric tumor cells [106]. Another study suggested that miR-221 in BMSC-derived exosomes, as a tumor-promoting molecule, promoted the proliferation and progression of gastric tumors by activating the Hedgehog signaling pathway [107]. MSCs can also promote immune evasion of tumor cells in the TME, and it was found that human umbilical cord mesenchymal stem cells (hUCMSC)-derived exosomes containing miR-3940-5p were able to downregulate EMT, metastasis and invasion of DLD-1 colorectal tumor cell line through decreasing the integrin α6 (ITGA6) expression and inhibiting the activity of TGF-β1 signaling pathway [108]. In addition, it was reported that miR-122-modified adipose MSC (AMSC)-derived exosomes could enhance the chemosensitivity of HCC cells [109]. In addition, Ma et al. showed that MSC-derived exosomes were electroporated with miR-132 mimics, and after co-culture with human umbilical vein endothelial cells (HUVECs), miR-132 was up-regulated in HUVECs and bound to the target gene RASA1, thereby promoting angiogenesis in myocardial infarction [110]. In another study, exosomes were isolated from adipose tissue-derived mesenchymal stem cells (AD-MSC-Exo), and then transfected into exosomes with miR-10a, and added to naive T cells. The secretion levels of IL-17 and TGF-β were increased, while the secretion level of IFN-γ was decreased, thus providing a new strategy for anti-tumor immunotherapy [111]. Therefore, researchers have investigated how to use exosomes as biological delivery vehicles for miRNA transfer. Therefore, MSC-derived exosomes can also be used as novel nanocarriers for miRNAs and drugs.

### 4.2. Macrophages-Derived Exosomes

Macrophages exhibit different phenotypes and functions under the stimulation of different factors in the surrounding environment [112]. IL-4-induced macrophages are selectively activated (M2) macrophages, and M2 macrophages are characterized by reduced secretion of proinflammatory cytokines and mannose receptors compared with IFN-γ-induced M1 macrophages [113]. It has been reported that the upregulation of miRNA-24-3p in M2 macrophage-derived exosomes reduced the expression of tumor necrosis factor superfamily member 10 (TNFSF10), thereby exerting a cardioprotective effect on myocardial injury after sepsis [114]. In addition, a large number of proteins in exosomes derived from LPS-treated macrophages are involved in the NOD-like receptor signaling pathway, and the NLRP3 inflammasome is also activated after hepatocytes ingested LPS-treated macrophage-derived exosomes, suggesting the importance of macrophage-derived exosomes in sepsis-induced liver injury [50]. The study by Jiang et al. (2019) found that miR-155 in serum exosomes of ALI mice increased the number of M1-type macrophages in the lung by targeting SHIP1 and SOCS1, and then causes lung inflammation [115]. A lot of evidence showes that macrophage-derived exosomes are the main source of early proinflammatory cytokines in severe ALI and may activate neutrophils to produce more proinflammatory cytokines and IL-10. IL-10 polarizes macrophages to M2 type, which in turn leads to subsequent fibrosis [116]. In addition, it was also reported that miR-590-3p in M2 macrophage-derived exosomes targeted LATS1 to activate YAP/β-catenin to attenuate inflammation and promote epithelial regeneration of damage mucosa from colitis mice induced by DSS [117].

We usually refer to macrophages in the tumor microenvironment (TME) as tumor-associated macrophages (TAMs). TAMs are also highly plastic and heterogeneous, accounting for 30–40% of immune cells in the TME [118]. Additionally, they are extravasated by circulating monocytes from nearby blood vessels and into tumor tissue, where they polarize into distinct phenotypes in the TME [119]. Most TAMs lack the ability to phagocytose tumor cells, and at the same time promote tumor cell immune evasion, allowing them to metastasize to distant tissues [120]. Exosomes secreted by TAMs also play a key role in regulating tumor progression. It regulates tumor progression by upregulating cancer proliferation, migration, invasion, promoting angiogenesis, generating drug resistance, promoting tumor immune escape and reprogramming tumor metabolic processes. For example, it was found that CD11b/CD18, an integrin derived from M2-type macrophage exosomes, promoted the metastasis of HCC cells by activating MMP-9 [121]. Another study found that exosomes derived from TAMs released large amounts of transcription factors GATA3, which played a key role in the interaction between TAMs and high-grade serious ovarian cancer (HGSOC), promoting the occurrence of EMT and angiogenesis [122]. TAMs-derived exosomes can also affect the immune response of different T cell subtypes in the TME, thereby promoting tumor cells to evade immune recognition [123].

### 4.3. Neutrophils-Derived Exosomes

Neutrophils are also the first line of defense in the innate immune response. Neutrophils can secrete exosomes that affect macrophages, endothelial cells, vascular and bronchial smooth muscle cells, etc. [124]. Extensive study has investigated the role of neutrophil-derived exosomes in regulating the local and systemic inflammatory environment. Neutrophils can also release exosomes and release IL-8 to play an anti-inflammatory role when they are not activated [125]. The exosomes released by stimulated neutrophils exhibited different properties. For example, exosomes released from neutrophils stimulated by TNF-α enhance the production of pro-inflammatory cytokines, leading to genomic instability, inflammation and impaired wound healing in intestinal epithelial cells [126]. Conversely, there are also reports of the anti-inflammatory effects of TNF-α-stimulated neutrophil-derived exosomes on a macrophage-fibroblast-like synovial cell co-culture system. It was reported that exosomes released from neutrophils induced by N-formyl-Met-Leu-Phe (fMLP) exhibited a pro-inflammatory phenotype when co-cultured with HUVEC and leading to the release of IL-8 and IL-6 [127]. On the other hand, it was found that exosomes released by fMLP-stimulated neutrophils interfered with NF-κB signaling in human monocyte-derived macrophages and not only promoted TGF-β1 release but also inhibited IFN-γ and TNF-α production, and thus neutrophil-derived exosomes showed significant anti-inflammatory effects [128]. MiRNA-30d-5p in neutrophil-derived exosomes induced M1-type macrophage polarization and triggered macrophage pyroptosis by activating NF-κB signaling, leading to sepsis-related ALI [129]. In contrast, during mechanically induced lung inflammation in mice, neutrophil-derived exosomes transferred miRNA-223 into alveolar epithelial cells and suppressed PARP-1 expression, thereby suppressing the deleterious inflammatory cascade during ALI [130]. This inconsistency may be due to differences in the stimuli that stimulate neutrophils to produce exosomes and the living environment of the studied target cells.

## 5. Concluding and Future Perspectives

The ability of inflammation to induce cancer has been well established over the past decade, but the mechanisms by which many inflammatory processes lead to tumor development have not been fully elucidated. Inflammation appears to drive all steps required for tumorigenesis, including angiogenesis, cell proliferation, migration and invasion and drug resistance [119,131]. Long-term chronic inflammation cause cancer to act like a “wound that will not heal” and produce an immunosuppressive TME. Inflammation is traditionally believed to be achieved by the interaction of various types of cells, directly or indirectly through the regulation of cytokines and other soluble factors [9]. More and more studies have demonstrated that exosomes released by various types of cells were also involved. Their effects can vary according to their source and the microenvironment in which the acting on target cells live in.

This article reviews the significance of exosomes in inflammatory diseases (Figure 1). In sepsis, exosomes carry a large number of pro-inflammatory molecules, activate inflammatory molecule-related signaling pathways and induce multiple organ dysfunction. Therefore, targeting excessively released exosomes is likely to be an effective treatment for sepsis. For example, increased activity of reduced coenzyme II (NADPH) in platelet-derived exosomes from patients with sepsis induces oxidative stress and triggers apoptosis in vascular epithelial cells [132]. When the early pathway for the production and release of exosomes was blocked, mice with sepsis had significantly improved survival and greatly attenuated myocardial damage [133]. This suggested that exosomes could be a novel target for sepsis therapy. In addition, blood culture is one of the most commonly used detection methods for the diagnosis of sepsis in clinical practice, but it is limited due to its low positive rate of culture and the long time required [134]. Therefore, researchers began to explore the use of exosomes in the body fluids of sepsis patients as an early detection indicator [135]. Exosomes are also ideal drug delivery vehicles in sepsis. Some scholars isolated dendritic cell-derived exosomes and filled them with exogenous milk fat globule epidermal growth factor VIII (MFGE8), a secreted protein necessary for the regulation and removal of apoptotic cells. The level of MFGE8 was significantly downregulated during the onset of toxicosis [136]. This suggested that exosomes had the ability to carry highly hydrophobic proteins such as MFGE8, enhancing their anti-inflammatory effects in sepsis. Exosomes also play a key role in various inflammatory diseases of the lungs, especially in lung cancer, where exosomes are involved in tumor metastasis, angiogenesis, immune escape and even drug resistance. Since tumor exosomes containing molecules of exosome-derived cells can be detected in the circulation, exosomes have been used as biomarkers to aid in the screening and early diagnosis of lung cancer, giving patients a better prognosis. For example, the clinical liquid biopsy kit developed by Exosome Diagnostics for analyzing exosomal RNA from blood samples was approved by the US FDA for clinical use in early 2016. It can accurately and real-time detect EML4-ALK mutations in non-small cell lung cancer (NSCLC) patients, and the detection can reach 88% diagnostic sensitivity and 100% diagnostic specificity. The study by Huang et al. (2013) found that 80% of NSCLC patients had positive EGFR immunostaining on the surface of exosomes in lung tissue, while only 2% of exosomes in chronic pneumonia were positive for EGFR. Therefore, it is considered that the exosomal EGFR protein could be used as a biomarker for the differential diagnosis of NSCLC and chronic pneumonia [137]. In a phase II clinical trial, Besse et al. (2016) found that dendritic cell-derived exosomes loaded with IFN-γ, MHC I and MHC II could enhance the anti-tumor immune function of NK cells in patients with advanced NSCLC [138]. The study by Wang et al. (2017) found that the use of exosomes to deliver paclitaxel (PTX) could significantly improve the absorption of PTX by lung cancer cells and significantly increase the cytotoxicity of the drug. Therefore, the encapsulation of PTX in exosomes could significantly inhibit the development of lung cancer [139]. In liver disease, stressed or damaged hepatocytes release large amounts of exosomes to promote inflammation and fibrogenesis. Additionally, due to the characteristics of exosomes, there are many opportunities for intervention in the formation of HCC, and it is very promising to find HCC-specific biomarkers. Lou et al. (2015) found that intraperitoneal injection of exosomes containing miR-122 secreted by adipose-derived mesenchymal stem cells (AMSCs) and sorafenib into mice could significantly improve the efficacy of sorafenib [109]. This reflects the important role of exosomes as drug delivery vehicles. Chronic intestinal inflammation leads to immune dysregulation and persistent destruction of IECs, and exosomes effectively modulate the barrier function of immune system cells, gut microbiota and IECs, show great potential as a new therapeutic modality. Additionally, exosomes have been reported as nanocarriers or CRC therapy targets with high specificity. Liu et al. (2019) packaged miR-128-3p into exosomes secreted by normal intestinal cells, and exosomes could effectively deliver miR-128-3p to oxaliplatin-resistant CRC cells, thereby improving the resistance response of CRC cells to oxaliplatin in vitro and in vivo [140].

Of course, there are still a large number of cells in the inflammatory environment that continuously release exosomes, which play a role in inflammatory diseases and even tumors caused by inflammation. Among them, MSCs-derived exosomes have shown outstanding application prospects in the treatment of sepsis, respiratory infections and intestinal infections. MSCs-derived exosomes can function through mRNA, miRNAs and proteins. In addition, there are a large number of patients with pneumonia who urgently need to treat COVID-19 infection, and exosomes from either MSCs or COVID-19-specific T cells may be one of the best treatments. Therefore, it may be more meaningful to develop exosomes as new vaccine and drug delivery systems. The surrounding inflammatory environment modulates the phenotype of macrophages, and thus the function of macrophage-derived exosomes is affected by the phenotype of macrophages. In the TME, the phenotype of TAMs is in dynamic changes, so the role of TAMs-derived exosomes is constantly changing. Since they can be easily modified by engineering, macrophage-derived exosomes can also be used as drug carriers with sufficient safety.

In addition, exosomes in the TME are constantly regulating tumor development. Tumor-derived exosomes contain components that promote or inhibit the further development of tumors. Existing studies have shown that most tumor-derived exosomes are involved in tumor cell proliferation, regulation of immune responses, regulation of epithelial-mesenchymal transition (EMT), tumor cells metastasis and plays a key role in angiogenesis (Figure 2).

Overall, as the understanding of cancer-related inflammation continues to increase, this knowledge is gradually being translated into new directions for tumor immunotherapy. In this process, increasing evidence supports the importance of exosomes in regulating inflammation and cancer-related inflammation. Understanding the TME at different stages of cancer progression can also provide a deeper understanding of the role of exosomes in the occurrence and development of tumors. At present, there is still a lack of in-depth and systematic research on the mechanism of exosomes in tumorigenesis and development in the inflammatory environment. For example, how tumor-derived exosomes and tumor cells or host cells recognize each other, how exosome content regulates tumor progression, and where inflammatory factors play a role need to be further studied. Further exploration is needed to expand the existing limited relationship between exosomes and inflammation or tumors. The basic and clinical translational research on exosomes and chronic inflammation-induced tumors should continue to be strengthened. Over the past few decades, research on exosomes has shown their great potential to improve the diagnosis and treatment of human diseases and exosomes are highly valuable targets in biomarker discovery, targeted drug delivery and vaccine development. However, it is still a challenge to find better methods to isolate and purify exosomes and to develop highly sensitive single tumor exosome detection tools. Therefore, the study of exosomes will provide more powerful guidance for the diagnosis and treatment of inflammatory diseases and tumors in the future.

## Figures and Tables

**Figure 1 cells-11-01005-f001:**
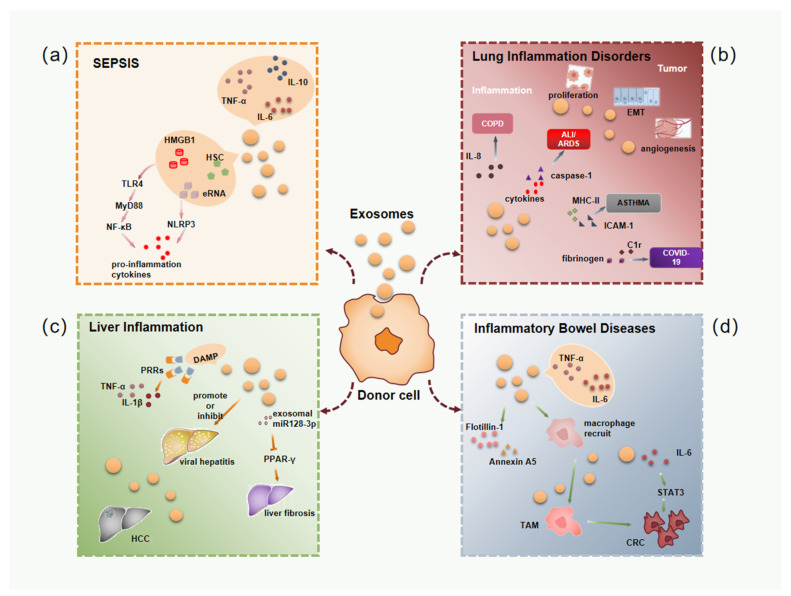
The role of exosomes in inflammatory diseases and tumor-related inflammation. (**a**) During sepsis, exosomes released by donor cells contain large amounts of inflammatory factors (TNF-α, IL-6, IL-10) and DAMP molecules, which lead to the activation of downstream inflammatory signaling pathways. (**b**) Exosomes lead to COPD, ALI/ARDS, ASTHMA and other pulmonary inflammatory diseases by encapsulating different molecules, such as IL-8, caspase-1, MHCII, ICAM-1, and may further promote cell proliferation, EMT and angiogenesis to promote the development of lung cancer. (**c**) DAMP molecules encapsulated by exosomes cause severe inflammatory responses in the liver. Exosomes derived from different donor cells also promote or inhibit the occurrence of viral hepatitis through different mechanisms. In addition, exosome-encapsulated miRNA-128-3p promote liver fibrosis by inhibiting PPAR-γ. Of course, exosomes also play an important role in the occurrence and development of liver cancer. (**d**) In intestinal inflammation, exosomes not only contain many inflammatory factors, but also induce the polarization of macrophages to TAM, both of which further lead to the occurrence and development of CRC. DAMP, damage-associated molecular pattern; COPD, chronic obstructive pulmonary disease; ALI/ARDS, acute lung injury/acute respiratory distress syndrome; EMT, epithelial-mesenchymal transition; TAM, tumor-associated macrophage; CRC, colitis-associated colorectal cancer.

**Figure 2 cells-11-01005-f002:**
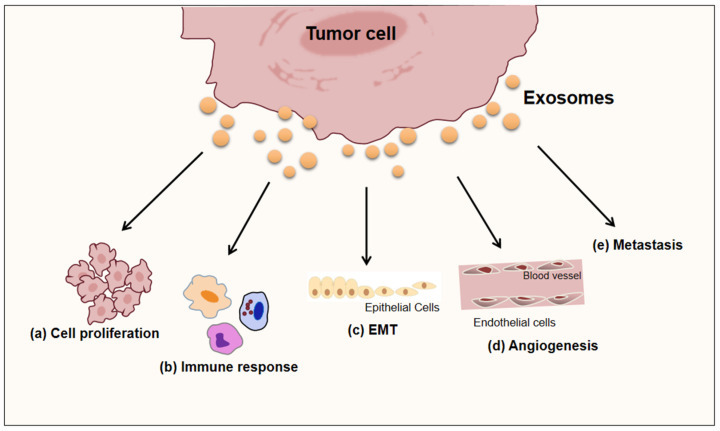
Tumor-derived exosomes promote tumor development. Tumor-derived exosomes contribute to the development of tumors via promoting tumor cell proliferation (**a**), regulating immune responses (**b**), enhancing epithelial-mesenchymal transition (EMT) (**c**) and angiogenesis (**d**), and strengthening tumor metastasis (**e**).

## Data Availability

Not applicable.

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
