# Peer review of "The Role of Exosomes in Inflammatory Diseases and Tumor-Related Inflammation"

_cells, 2022, doi:10.3390/cells11061005_

Round 1

Reviewer 1 Report

This is an interesting review article. The authors have given an overview of exosomes and its role in inflammation and cancer.

Please find my comments as bellow.

MAJOR:

In the abstract, the authors have stated about discussing (in-depth) the potential of exosomes as targets and tools to treat inflammatory diseases and tumor-related inflammation. Unfortunately, I could not find any section related to it.

MINOR:

(1)  Please proofread the whole manuscript for grammatical errors. Please ensure the correct use of present and past tense.

(2)  Page 1; Line 40,41: increased genomic instability, abnormal cell proliferation, changes in the stromal environment and transitions between epithelial and mesenchymal states do not represent multiple cellular interactions. Please change.

(3)      Page 1; line 42: No earlier mention of cytokines.

(4)  The full form of abbreviation TAM has been repeated more than twice. Please maintain consistency. Please provide the full form of APAP (Page 4; line 179).

(5)  Page 2; line 90: Please correct the full form of MVB.

(6)  Reference 38 is not a recent article. Cite the recent article by Urabe et al. (2021) describing the role of extracellular vesicles in metastasis: Urabe F, Patil K, Ramm GA, Ochiya T, Soekmadji C. Extracellular vesicles in the development of organ-specific metastasis. J Extracell Vesicles. 2021 Jul;10(9):e12125. doi: 10.1002/jev2.12125. Epub 2021 Jul 19. PMID: 34295457; PMCID: PMC8287318.

(7)  Page 3; lines 109-112: The lines convey the same meaning. Please reframe to avoid redundancy.

(8)  There are several sentences that begin with ‘the study’. If you wish to begin with it, either you mention it is as ‘the study by … et al. (year)’ or begin with as ‘one study’.

(9) Page 3; lines 124-127: Please reframe the sentence for better understanding.

(10) Page 5; line 205: Do the authors mean exosomes are involved in the development of chronic hepatitis B (CHB) and chronic hepatitis C (CHC) infections that are the causative factors of  HCC ?

(11) Page 5; line 237: Do the authors mean infiltration of inflammatory cells or factors?

(12) In my opinion, heading 4 should be changed to ‘Role of exosomes released by microenvironmental cells in inflammatory diseases and tumor-associated inflammation’.

(13) Avoid starting the sentence with ‘And’.

(14) Page 7; line 326: a large number of transcription factors should be replaced by large amounts of.

(15) Ref 123 does not belong to the section on macrophage-derived exosomes. Please check again.

(16) Kindly correct the spellings in the figure such as in the box related to inflammation, the spelling of inflammation is incorrect.

(17) A short figure legend describing the pathways will be good.

(18) Page 4; line 176: ‘In contrast’ should be used carefully. It has been used in places that are unnecessary throughout the manuscript. Please delete wherever it is not suitable.

(19) Page 6; lines 289-291: Kindly cite appropriate papers and mention a few examples to justify the statement ‘Therefore, researchers have investigated how to use exosomes as biological delivery vehicles for miRNA transfer.’

(20) Page 6; lines 301-305: Please reframe or split the sentence for a better understanding.

(21) The language style of the last paragraph of the concluding and future perspectives does not match with the rest of the manuscript. It is well-written and the style should be followed throughout.

(22) Please check whether ref 103 belongs to the section on macrophage-derived exosomes.

Author Response

Response to Reviewer 1 Comments

MAJOR:

In the abstract, the authors have stated about discussing (in-depth) the potential of exosomes as targets and tools to treat inflammatory diseases and tumor-related inflammation. Unfortunately, I could not find any section related to it.

Response: Thank you very much for your comments. We have added to the new manuscript about the potential of exosomes as a treatment for inflammatory diseases and tumor-associated inflammation. In the second paragraph of the ‘Concluding and future perspectives’, we have added discussions about the role of exosomes as targets and tools to treat inflammatory diseases.

We added ‘For example, increased activity of reduced coenzyme II (NADPH) in platelet-derived exosomes from patients with sepsis induces oxidative stress and triggers apoptosis in vascular epithelial cells[132]. When the early pathway for the production and release of exosomes was blocked, mice with sepsis had significantly improved survival and greatly attenuated myocardial damage[133]. This suggested that exosomes could be a novel target for sepsis therapy. In addition, blood culture is one of the most commonly used detection methods for the diagnosis of sepsis in clinical practice, but it is limited due to its low positive rate of culture and the long time required[134]. Therefore, re-searchers began to explore the use of exosomes in the body fluids of sepsis patients as an early detection indicator[135]. Exosomes are also ideal drug delivery vehicles in sepsis. Some scholars isolated dendritic cell-derived exosomes and filled them with exogenous milk fat globule epidermal growth factor VIII (MFGE8), a secreted protein necessary for the regulation and removal of apoptotic cells. The level of MFGE8 was significantly down-regulated during the onset of toxicosis[136]. This suggested that exosomes had the ability to carry highly hydrophobic proteins such as MFGE8, en-hancing their anti-inflammatory effects in sepsis.‘ on page 8, line 387-402 of the new manuscript.

We added ‘For example, the clinical liquid biopsy kit developed by Exosome Diagnostics for ana-lyzing exosomal RNA from blood samples was approved by the US FDA for clinical use in early 2016. It can accurately and real-time detect EML4-ALK mutations in non-small cell lung cancer (NSCLC) patients, and the detection can reach 88% diag-nostic sensitivity and 100% diagnostic specificity. The study by Huang et al. (2013) found that 80% of NSCLC patients had positive EGFR immunostaining on the surface of exosomes in lung tissue, while only 2% of exosomes in chronic pneumonia were pos-itive for EGFR. Therefore, it is considered that the exosomal EGFR protein can be used as a biomarker for the differential diagnosis of NSCLC and chronic pneumonia[137]. In a phase II clinical trial, Besse et al. (2016) found that dendritic cell-derived exosomes loaded with IFN-γ, MHC I and MHC II could enhance the anti-tumor immune func-tion of NK cells in patients with advanced NSCLC[138]. The study by Wang et al. (2017) found that the use of exosomes to deliver paclitaxel (PTX) could significantly improve the absorption of PTX by lung cancer cells and significantly increase the cyto-toxicity of the drug. Therefore, the encapsulation of PTX in exosomes could signifi-cantly inhibit the development of lung cancer[139].’ on page 8-9, line 407-421 of the new manuscript.

We added ‘‘Lou et al. (2015) found that intraperitoneal injection of exosomes containing miR-122 secreted by adipose-derived mesenchymal stem cells (AMSCs) and sorafenib into mice could significantly improve the efficacy of sorafenib[109]. This reflects the important role of exosomes as drug delivery vehicles.’ on page 9, line 425-428 of the new manuscript.

We added ’Liu et al. (2019) packaged miR-128-3p into exosomes secreted by normal intestinal cells, and exosomes could effectively deliver miR-128-3p to oxaliplatin-resistant CRC cells, thereby improving the resistance response of CRC cells to oxaliplatin in vitro and in vivo[140].’ on page 9, line 432-435 of the new manuscript.

MINOR:

(1) Please proofread the whole manuscript for grammatical errors. Please ensure the correct use of present and past tense.

Response 1:Thanks. We have re-proofed the entire manuscript for grammatical errors.

(2) Page 1; Line 40,41: increased genomic instability, abnormal cell proliferation, changes in the stromal environment and transitions between epithelial and mesenchymal states do not represent multiple cellular interactions. Please change.

Response 2:Thanks. .We have corrected this sentence into ’Inflammation-promoting carcinogenesis may be the result of a changing environment interacting with a variety of cells, such as increased genomic instability, abnormal cell proliferation, changes in the stromal environment, and transitions between epithelial and mesenchymal states’.

(3) Page 1; line 42: No earlier mention of cytokines.

Response 3: Thanks. In order to describe more accurately, we have changed ‘Cytokines’ to ‘Inflammatory factors’.

(4) The full form of abbreviation TAM has been repeated more than twice. Please maintain consistency. Please provide the full form of APAP (Page 4; line 179).

Response 4:Thanks. We have removed page 5, line 252 about the full form of abbreviation TAM and added the full form of APAP, which is acetaminophen.

(5) Page 2; line 90: Please correct the full form of MVB.

Response 5:Thanks. We have corrected the full form of MVB to multivesicular body.

(6) Reference 38 is not a recent article. Cite the recent article by Urabe et al. (2021) describing the role of extracellular vesicles in metastasis: Urabe F, Patil K, Ramm GA, Ochiya T, Soekmadji C. Extracellular vesicles in the development of organ-specific metastasis. J Extracell Vesicles. 2021 Jul;10(9):e12125. doi: 10.1002/jev2.12125. Epub 2021 Jul 19. PMID: 34295457; PMCID: PMC8287318.

Response 6:Thanks. We have replaced reference 38 with Urabe F, Patil K, Ramm GA, Ochiya T, Soekmadji C. Extracellular vesicles in the development of organ-specific metastasis. J Extracell Vesicles. 2021 Jul;10(9):e12125.

(7) Page 3; lines 109-112: The lines convey the same meaning. Please reframe to avoid redundancy.

Response 7:Thanks. We have deleted the phrase ‘Inflammation in sepsis can be mediated by a variety of signals, of which exosomes also play an important role’ to avoid redundancy.

(8) There are several sentences that begin with the study. If you wish to begin with it, either you mention it is as the study by et al. (year) or begin with as one study.

Response 8:Thanks. We changed ‘The study’ on page 3, line 118 to ‘One study’, and ‘The study’ on page 3, line 148 to ‘The study by Lee et al. (2018)’, ‘The study’ on page 7 line 313 is changed to ‘The study by Jiang et al. (2019)’.

(9) Page 3; lines 124-127: Please reframe the sentence for better understanding.

Response 9:Thanks. We have reframed the sentence and changed Page 3, lines 124-127 to 'The most well-known mechanism of sepsis is that lipopolysaccharide (LPS) activates the TLR4-MyD88 pathway, which further activates the downstream NF-κB signaling pathway, resulting in the production of a large number of inflammatory molecules.'

(10) Page 5; line 205: Do the authors mean exosomes are involved in the development of chronic hepatitis B (CHB) and chronic hepatitis C (CHC) infections that are the causative factors of HCC ?

Response 10:Thanks. We found in references that exosomes are involved in the development of chronic hepatitis B (CHB) and chronic hepatitis C (CHC) infections. HCC is a major complication of CHB and CHC.

(11) Page 5; line 237: Do the authors mean infiltration of inflammatory cells or factors?

Response 11:Thanks. Yes, it means infiltration of inflammatory cells.

(12) In my opinion, heading 4 should be changed to Role of exosomes released by microenvironmental cells in inflammatory diseases and tumor-associated inflammation.

Response 12:Thanks. We have changed heading 4 to Role of exosomes released by microenvironmental cells in inflammatory diseases and tumor-associated inflammation.

(13) Avoid starting the sentence with And.

Response 13:Thanks. We have deleted the 'And' on page 1, line 26 , 71 and page 4, line 165.

(14) Page 7; line 326: a large number of transcription factors should be replaced by large amounts of.

Response 14:Thanks. We have replaced ‘a large number of’ with ‘large amounts of’ on page 7, line 326.

(15) Ref 123 does not belong to the section on macrophage-derived exosomes. Please check again.

Response 15:Thanks. We have deleted Ref 123.

(16) Kindly correct the spellings in the figure such as in the box related to inflammation, the spelling of inflammation is incorrect.

Response 16:Thanks. We have corrected the spelling of inflammation and the new figure 1 is as follows:

(17) A short figure legend describing the pathways will be good.

Response 17:Thanks. We have added a short figure legend to describe the pathways and now the figure legend is:

Figure 1. The role of exosomes in inflammatory diseases and tumor-related inflammation.

  • During sepsis, exosomes released by donor cells contain large amounts of inflammatory factors (TNF-α, IL-6, IL-10) and DAMP molecules, which lead to the activation of downstream inflammatory signaling pathways. (b) Exosomes lead to COPD, ALI/ARDS, ASTHMA and other pulmonary inflammatory diseases by encapsulating different molecules, such as IL-8, caspase-1, MHCII, ICAM-1, and may further promote cell proliferation, EMT and angiogenesis to promote the development of lung cancer. (c) DAMP molecules encapsulated by exosomes cause severe inflammatory responses in the liver. Exosomes derived from different donor cells also promote or inhibit the occurrence of viral hepatitis through different mechanisms. In addition, exosome-encapsulated miRNA-128-3p promote liver fibrosis by inhibiting PPAR-γ. Of course, exosomes also play an important role in the occurrence and development of liver cancer. (d) In intestinal inflammation, exosomes not only contain many inflammatory factors, but also induce the polarization of macrophages to TAM, both of which further lead to the occurrence and development of CRC.

(18) Page 4; line 176: In contrast should be used carefully. It has been used in places that are unnecessary throughout the manuscript. Please delete wherever it is not suitable.

Response 18:Thanks. We have deleted ‘In contrast’ on page 4, line 176.

(19) Page 6; lines 289-291: Kindly cite appropriate papers and mention a few examples to justify the statement Therefore, researchers have investigated how to use exosomes as biological delivery vehicles for miRNA transfer.

Response 19:Thanks. We have cited two examples to demonstrate ‘Therefore, researchers have investigated how to use exosomes as biological delivery vehicles for miRNA transfer’. The content is ‘In addition, Ma et al. showed that MSC-derived exosomes were electroporated with miR-132 mimics, and after co-culture with HUVECs, miR-132 was up-regulated in HUVECs and bound to the target gene RASA1, thereby promoting angiogenesis in myocardial infarction. In another study, miR-10a was introduced into AD-MSC-Exo, exosomes isolated from adipose tissue-derived mesenchymal stem cells, and then delivered by AD-MSC-Exo into naive T cells, which results in the secretion levels of IL-17 and TGF-β are increased, while the secretion level of IFN-γ is decreased, thus providing a new strategy for anti-tumor immunotherapy.’

(20) Page 6; lines 301-305: Please reframe or split the sentence for a better understanding.

Response 20:Thanks. We have reframed this sentence into ‘IL-4-induced macrophages are selectively activated (M2) macrophages. Compared to IFN-γ-induced M1 macrophages, M2 macrophages are characterized by the decreased secretion of proinflammatory cytokines and mannose receptors’.

(21) The language style of the last paragraph of the concluding and future perspectives does not match with the rest of the manuscript. It is well-written and the style should be followed throughout.

Response 21:Thanks. We have tried to match the rest of the article with the style of the last paragraph as you suggested.

  • Please check whether ref 103 belongs to the section on macrophage-derived exosomes.

Response 22:Thanks. We have corrected ref 103 to ‘Wang, J., R. Huang, Q. Xu, G. Zheng, G. Qiu, M. Ge, Q. Shu, and J. Xu. "Mesenchymal stem cell-derived extracellular vesicles alleviate acute lung injury via transfer of Mir-27a-3p." Crit Care Med 48, no. 7 (2020): e599-e610.’.

Reviewer 2 Report

The authors present a review of the potential roles of exosomes in inflammatory diseases and tumor-associated inflammation. The review is well presented, gives a nice overview of exosomes and the evidence for their roles in inflammation and some tumor biology, illustrated with a diagram with interactions of exosomes with several inflammatory diseases.

The figure presented shows potential roles of exosomes and the effect on tumorigenesis, COVID-19 and sepsis. The legend should be expanded on, explaining in more detail each of the panels. Another figure or additional panel in the current figure specifically detailing tumor-derived exosomes and their potential effects eg. metastasis, local immune signaling, EMT, angiogenesis etc. would enhance the manuscript. 

Author Response

Response to Reviewer 2 Comments

The figure presented shows potential roles of exosomes and the effect on tumorigenesis, COVID-19 and sepsis. The legend should be expanded on, explaining in more detail each of the panels. Another figure or additional panel in the current figure specifically detailing tumor-derived exosomes and their potential effects eg. metastasis, local immune signaling, EMT, angiogenesis etc. would enhance the manuscript.

Response:Thank you very much for your comments. We have added a short figure legend to describe pathways and now the figure and figure legend are as follows:

Figure 1. The role of exosomes in inflammatory diseases and tumor-related inflammation.

  • During sepsis, exosomes released by donor cells contain large amounts of inflammatory factors (TNF-α, IL-6, IL-10) and DAMP molecules, which lead to the activation of downstream inflammatory signaling pathways. (b) Exosomes lead to COPD, ALI/ARDS, ASTHMA and other pulmonary inflammatory diseases by encapsulating different molecules, such as IL-8, caspase-1, MHCII, ICAM-1, and may further promote cell proliferation, EMT and angiogenesis to promote the development of lung cancer. (c) DAMP molecules encapsulated by exosomes cause severe inflammatory responses in the liver. Exosomes derived from different donor cells also promote or inhibit the occurrence of viral hepatitis through different mechanisms. In addition, exosome-encapsulated miRNA-128-3p promote liver fibrosis by inhibiting PPAR-γ. Of course, exosomes also play an important role in the occurrence and development of liver cancer. (d) In intestinal inflammation, exosomes not only contain many inflammatory factors, but also induce the polarization of macrophages to TAM, both of which further lead to the occurrence and development of CRC.

We also added a new figure about tumor-derived exosomes and their potential effects.

Figure 2. Tumor-derived exosomes promote tumor development. Tumor-derived exosomes contribute to the development of tumors via promoting tumor cell proliferation (a), regulating immune responses (b), enhancing epithelial-mesenchymal transition (EMT) (c), and angiogenesis (d), and strengthening tumor metastasis (e).

Round 2

Reviewer 1 Report

well responded